# Neural Indicators of Visual Andauditory Recognition of Imitative Words on Different De-Iconization Stages

**DOI:** 10.3390/brainsci13040681

**Published:** 2023-04-19

**Authors:** Liubov Tkacheva, Maria Flaksman, Yulia Sedelkina, Yulia Lavitskaya, Andrey Nasledov, Elizaveta Korotaevskaya

**Affiliations:** 1Department of Pedagogy and Pedagogical Psychology, Saint Petersburg State University, 199034 Saint Petersburg, Russia; 2Department for English and American Studies, Ludwig Maximilian University, 80799 München, Germany; 3Department of Foreign Languages and Linguo-Didactics, Saint Petersburg State University, 199034 Saint Petersburg, Russia; 4Department of Clinical Psychology, Saint Petersburg State University, 199034 Saint Petersburg, Russia

**Keywords:** EEG, ERP, de-iconization, iconicity, Russian language

## Abstract

The research aims to reveal neural indicators of recognition for iconic words and the possible cross-modal multisensory integration behind this process. The goals of this research are twofold: (1) to register event-related potentials (ERP) in the brain in the process of visual and auditory recognition of Russian imitative words on different de-iconization stages; and (2) to establish whether differences in the brain activity arise while processing visual and auditory stimuli of different nature. Sound imitative (onomatopoeic, mimetic, and ideophonic) words are words with iconic correlation between form and meaning (iconicity being a relationship of resemblance). Russian adult participants (*n* = 110) were presented with 15 stimuli both visually and auditorily. The stimuli material was equally distributed into three groups according to the criterion of (historical) iconicity loss: five explicit sound imitative (SI) words, five implicit SI words and five non-SI words. It was established that there was no statistically significant difference between visually presented explicit or implicit SI words and non-SI words respectively. However, statistically significant differences were registered for auditorily presented explicit SI words in contrast to implicit SI words in the N400 ERP component, as well as implicit SI words in contrast to non-SI words in the P300 ERP component. We thoroughly analyzed the integrative brain activity in response to explicit IS words and compared it to that in response to implicit SI and non-SI words presented auditorily. The data yielded by this analysis showed the N400 ERP component was more prominent during the recognition process of the explicit SI words received from the central channels (specifically Cz). We assume that these results indicate a specific brain response associated with directed attention in the process of performing cognitive decision making tasks regarding explicit and implicit SI words presented auditorily. This may reflect a higher level of cognitive complexity in identifying this type of stimuli considering the experimental task challenges that may involve cross-modal integration process.

## 1. Introduction

The phenomenon of iconicity in light of systematic and interdisciplinary approach has triggered increasing interest among researchers in recent years [1]. Iconicity is a relationship of resemblance between the form of a word and its meaning [2]. Language iconicity manifests itself in imitative (onomatopoeic, sound-symbolic) words and ideophones [3,4]. Sound imitative (SI) words share a considerable degree of cross-linguistic similarity (e.g., cf. the following denotations of a cat’s cry—Rus. mjau, Eng. meow, Vie. miu miu, Chi. (Mandarin) miāo). It is, thus, believed that iconicity is a design feature of the human language. Indeed, words with at least some degree of iconicity are found in all known languages across the globe [5].

Iconicity is also known for its facilitating effect on learning new words [6] and, according to the sound–symbolism bootstrapping hypothesis, it can also be considered as a bridge between learning SI words and other aspects of language [7]. It is assumed that the iconic mechanisms link lexical units with the cognitive representation of sensorimotor sensations that reflect physical reality. As suggested, these mechanisms contribute to the assimilation and processing of lexical meaning [8]. It was also suggested that the process of language acquisition in early childhood involves establishing links between similar entities and relevant iconic sounds or objects [9]. It is assumed that, eventually, the learned SI words can be generalized and replaced with other, less iconic types of words [10]. It is also hypothesized that iconicity has played an important role in the origin and evolution of language [11,12,13,14].

Currently, we have no direct means of studying the hypothetic proto-language of the ancestors of the *Homo sapiens* species in vivo. Thus, there are no empirical means of establishing the role of lexical iconicity in the origin and evolution of language. However, there is a possibility of conducting research on imitative words existing in modern languages in order to study their evolution and discover neurophysiological mechanisms underlying their perception and recognition. In our previous research, we studied visual recognition of imitative words by means of the lexical decision task [15]. We learnt that visually presented Russian and English SI words are recognized differently by native speakers of these languages in terms of speed, accuracy, and errors of recognition in comparison with the non-SI ones. This phase of our research was preceded by the stage of selection and validation of the lexical stimuli according to the criterion of iconicity loss. Iconicity loss (the degree of de-iconization) is established by the method of diachronic evaluation of the imitative lexicon [13].

SI words change over time. Language change (both phonetic and semantic) affects the iconic form–meaning link in a word. Gradual form transformations and semantic ramifications (taking place in every living language on a regular basis), therefore, make SI words less iconic over time. Flaksman [14] identified that imitative words existing simultaneously in present-day languages may be classified into four categories (or de-iconization stages (SDs)). In the course of (natural) language evolution the most ‘vivid’ iconic interjections on SD-1 (ha-ha!, zzz!) are being transformed to words on SD-4 (barbarian, rumor), which have completely lost the original form–meaning resemblance under the influence of (regular) sound changes and semantic shifts. The suggested criteria for the classification of imitative words according to de-iconization stages are the following: (1) morphological and syntactic integration; (2) presence of (regular) sound changes; and (3) presence of semantic shifts which lead to the loss of the original (sound-related) meaning [14].

In general, the stage of de-iconization of an iconic word, from our point of view, is an important parameter that should be always taken into account when conducting any experiments with imitative vocabulary. Since imitative words are (objectively) iconic to varying degrees, the use of SI words at different de-iconization stages in experimental research on, for example, perception of iconic vocabulary, yields different results (since, from the point of view of diachronic analysis, words at different SDs are words from different strata of the language’s evolutionary development). Thus, our current EEG research on neural indicators of visual and auditory recognition of imitative words is based on the concept of de-iconization, which is in line with our previous studies [15,16,17,18].

We predict the existence of at least two neurophysiological mechanisms behind the process of iconic words’ perception: (1) cross-modality, which provides an expected match between two or more parameters or aspects from different sensory modalities (for example, between such parameters as brightness and volume) [16]; and (2) synesthesia, which is responsible for resemblance-based mapping between the form and meaning of an iconic word [17]. Many studies have aimed to discover neural indicators of auditory recognition of iconic words with the use of EEG, ERP, and MRI. It was found that early negative EEG waveforms indicated a susceptibility to sound–symbolic label–object associations in children at the pre-speech stage of their development [18]. Those results are correspondent to another EEG study conducted on 11-month-old infants. The authors revealed the presence of large-scale synchronization in the left hemisphere, which was sensitive to the association of sound and symbol. They concluded that children at the preverbal stage of their development are already able to associate an auditory stimulus with visual perception by involving a multimodal information processing system [19]. The results of EEG study with registration of ERP showed how participants assimilated foreign iconic words in a state of congruence (the word and its translation coincide) and in a state of non-congruence (the translation does not correspond to the word). Participants were significantly better at identifying words in a state of congruence. The analysis of ERP showed that when perceiving words in the congruence condition, a larger P3 component and late positive complex (LPC) were registered than when perceiving words in the non-congruence condition. The authors believe that cross-modal correspondences between sound and meaning facilitate word learning, while cross-modal inconsistencies make this difficult, especially for people who are more sensitive to sound symbolism [20]. In another EEG experiment with ERP registration during the process of reading sentences by native Japanese speakers, the authors found that iconic words evoked a larger P2 component and a larger LPC compared to arbitrary words. They argue that P2 reflects the multisensory integration of sounds and related sensory representations, while that LPC may indicate higher requirements for cognitive processing of iconic words [21]. Several neuroimaging studies have shown the unique contribution of the superior temporal sulcus to the processing of iconic words [22,23]. According to MRI studies, the neural activity underlying iconic words processing might co-localize with activity related to multisensory integration, e.g., in the superior temporal sulcus by audiovisual synchrony [24], or in the intraparietal sulcus when audiovisual spatial congruency is involved [25]. These results partially support the hypothesis that iconic word-processing may include sensory-dependent and sensory-independent neural networks in the brain. However, since these results were limited to a comparison between sound symbols and visually represented objects, the processing of visual information may also involve mechanisms dependent on sensory modality. However, there is a research gap in the field concerning neural indicators of visual recognition of iconic words. Moreover, to the best of our knowledge, no studies have yet investigated the integrative brain activity in the process of visual and auditory recognition of iconic words classified according to stages of de-iconization.

The aim of this research was to study reorganization of integrative brain activity in the process of visual and auditory recognition of explicit iconic words, implicit iconic words, and non-iconic words. We supposed that the main differences between the recognition of the stimuli of these types will be reflected in long latency ERP (the P300, the N400), since they are considered as an objective measure of cognitive processes such as discrimination, memory, attention, and detection of stimuli [26]. We also expected to register a specific brain response from the premotor cortex (central EEG channels) on explicit SI words regardless of the type of presentation (visual or auditory) because all explicit iconic words are verbs denoting movement. Furthermore, according to the idea of cross-modal correspondences [27], which could be described as congruency effects between perceived stimuli [28], we expected to register associations between the iconic stimuli addressing different sensory modalities as neural indicators of multisensory processing. This could result in mapping between sensory cues in correspondent regions of the brain. Based on the results of the previous studies, we introduce the following hypotheses:

**Hypothesis** **1 (H1).**
*The recognition of explicit iconic words, implicit iconic words, and non-iconic words presented auditorily results in distinguishable differences in the integrative brain activity reflected in the long latency ERP.*


**Hypothesis** **2 (H2).**
*The recognition of explicit iconic words, implicit iconic words, and non-iconic words presented visually results in distinguishable differences in the integrative brain activity reflected in the long latency ERP.*


**Hypothesis** **3 (H3).**
*Recognition of explicit iconic words regardless of the type of stimuli presentation (audial or visual) may result in cross-modal correspondence, which may involve the specific long latency ERP component from central leads.*


## 2. Materials and Methods

### 2.1. Research Material

The stimuli material for the research were 15 Russian words equally distributed into 3 groups according to the criterion of iconicity loss. Those words were selected out of 64 semantic stimuli scrupulously drawn from The Russian Etymological Dictionary by Max Vasmer [29], The Dictionary of Russian Phonosemantic Abnormalities by Shliahova [30]. Only those words that met the pre-defined clear-cut criteria of homomorphism (in terms of the word’s length (monosyllabic), the lexical category, and the mean frequency of the groups of stimuli [31]) were chosen. The criterion for the 15 stimuli selection for the EEG experiment was that those stimuli were “typical” representatives of their groups (explicit SI words, implicit SI words, and non-SI words). They were adjusted in terms of speed, accuracy, and identification errors measured by means of the lexical decision task in our previous study [15,32]. To investigate the degrees of iconicity in word recognition, the experiment by Sidhu et al. [33] was partly replicated. Lexical stimuli selected for EEG experiment are presented in Table 1. Visual lexical stimuli were designed in the form of an inscription in black letters on a light background. Auditory lexical stimuli were recorded at a recording studio, voiced by female and male voices to control the announcer’s voice gender’s influence in the EEG experiment.

The main experiment was preceded by a preparatory step, which included the selection and validation of visual stimulus material for each lexical stimulus in accordance with the developed design of the study. In order to validate these visual stimulus materials, 3 images were selected for each semantic stimulus and the expertise was carried out (*n* = 50). As a result of this expertise, only those visual stimuli (1 out of 3 options) which were unanimously chosen by the experts were used for the EEG experiment. We then used test.psy software to generate experimental session and present lexical stimuli (visually and auditorily) automatically and in random order, followed by 2 visual stimuli for categorization, one of which was always congruent to the target stimuli (corresponding in the meaning) while the another was non-congruent.

### 2.2. Participants

110 Russian adult participants, who were native speakers of the Russian language and constituted 50 men and 60 women aged 18–45 (M = 23), took part in the experiment after signing up the informed consent officially approved by ethical committee of Saint-Petersburg State University. We used the following eligibility criteria to recruit participants and avoid possible bias: no bilingual speakers, no individuals with visual and hearing impairments, no individuals with a history of neurological or psychiatric diseases, and no individuals using any medication affecting speed of reaction, attention, perception, memory, or other drugs that in the opinion of the research team may affect the results of the study. Each subject was financially rewarded with ₽1000 upon completion of the task. After receiving the financial reward, the participants provided a receipt for the received remuneration.

### 2.3. Procedure

The experiment was conducted in a shielded chamber with sound insulation; the experiment leader was in an adjacent room, controlling the process of EEG data recording. The EEG study included recording visual and auditory ERP in response to visual and auditory presentation of the selected lexical stimuli and the subsequent categorization of visual stimuli material. We used this experimental design in order to register auditory and visual ERP on 3 types of lexical stimuli of varying degrees of iconicity. The task of visual stimuli material categorization was given to the participants in order to: (1) check the correctness of target stimuli recognition; and (2) increase the complexity of the experimental task involving the cognitive process of categorization as an analogue of a lexical decision task. Each participant was informed about the experimental conditions and gave their written informed consent prior to participating in the study in compliance with the Declaration of Helsinki; they also reported that they had not taken medications that could potentially affect their reaction time, had normal or adjusted to normal vision, and did not experience mental, psychiatric, or neurological disorders.

19 monopolar leads were arranged symmetrically according to the International 10–20 System (Fp1, Fp2, Fp3, Fp4, F7, F8, C3, C4, Fz, Cz, Pz, T3, T4, T5, T6, P3, P4, O1, and O2). An electrooculogram (EOG) was recorded to register eye movements artifacts by using 4 vertical and 2 horizontal channels. The averaged potential of two ear clip electrodes was used as reference. EEG was recorded continuously both in the background state (quiet wakefulness with the eyes closed and opened) and while the subjects were performing the tests on semantic stimuli recognition and visual stimuli categorization. At the beginning of observation, EEG was recorded with eyes closed and opened (1 min each). A subject then received the instructions (on a monitor screen and through speakers) explaining the task sequence. The objective was to recognize a semantic stimulus as soon as possible and confirm the decision by pressing the key button corresponding to the picture congruent to the semantic stimulus. There was no time constraint on the choice of a visual stimulus matching the lexical one.

In total, each participant was presented with 60 words (30 visually and 30 auditorily). The task of the subjects was to identify the word meaning by choosing the congruent picture out of two (corresponding by meaning to the lexical stimulus) and pressing the right or left arrow key (corresponding to the right or left picture respectively). As soon as the subject indicated their choice of visual stimulus by pressing the corresponding button on the keyboard, they were presented with the next lexical stimulus audially or visually in random order so as to control the possible interfering variable of stimuli displaying sequence. EEG was recorded using a Mitsar electroencephalograph and WinEEG software, with a 250 Hz sampling rate for each channel and a trial length of 1000 ms after stimulus onset. EEG data was filtered by setting the appropriate passbands in WinEEG from 0.5 Hz to 70 Hz in order to avoid low- and high-frequency noise, such as heart signals related to pulse artifacts, electromagnetic field noise, and interferences of lamps and devices on EEG signals.

### 2.4. Data Analysis

The main data analysis was preceded by the artefacts removal procedure. The following steps were taken to extract muscle and eye movement artefacts from EEG data: (1) using WinEEG software, an Independent Component Analysis (ICA) was carried out; (2) the time course of every ICA component was correlated with each of the vertical (4 channels) and horizontal (2 channels) EOG channel time courses; (3) those ICA components that highly correlated with one or more EOG channel time courses were removed from the EEG data; (4) epochs containing artefacts in one or more channel, along with noisy channels, were identified and removed from further analysis.

The visual and auditory ERP were calculated using Repeated Measures Analysis of Variance (ANOVA); the Post Hoc pairwise comparison criterion with Bonferroni’s correction was then applied to study the combined influence of the factors for explicit and implicit iconic words and non-iconic words on integrative brain activity in the process of their visual and auditory processing to determine the significance of differences between conditions. All the statistical data analyses were performed in STATISTICA software version 10.

Subsequently, we compared in more detail the changes of the integrative brain activity in response to explicit SI words and non-SI words presented auditorily on all 19 EEG using WinEEG software developed by Mitsar. The t-Student criterion was used with the Tukey correction for the multiplicity of comparisons. The visual and auditory ERP were again calculated using Repeated Measures Analysis of Variance (ANOVA) for Cz channel and time windows correspondent to the long latency ERP (time window of 375–425 ms) in order to check our third hypothesis concerning cross-modal correspondence involving premotor cortex, while recognizing explicit iconic words regardless of the type of stimuli presentation (auditorily or visually).

## 3. Results

The results of the Repeated Measures Analysis of Variance are presented in Table 2.

The integrative activity of the brain significantly differs in the process of visual and auditory recognition of SI words depending on the modality and time parameters of integrative brain activity.

The influence of modality (visual, auditory) on the time parameters in integrative activity of the brain is presented in Figure 1.

The results of the Post Hoc analysis with Bonferroni correction significance of differences between conditions are presented in Table 3.

It was established that there was no statistically significant difference between visually presented explicit and implicit SI words and non-SI words. However, statistically significant differences were obtained for auditorily presented explicit SI words in contrast to the implicit SI words in the N400 ERP component (*p* = 0.014050), as well as for the implicit SI words in contrast to non-SI words in the P300 ERP component (*p* = 0.043261).

The differences between three groups of stimuli (explicit and implicit SI words and non-SI words) in both modalities in terms of the time parameters in integrative activity of the brain are presented in Figure 2.

Next, we decided to compare in more detail the changes in integrative brain activity in response to explicit SI words and non-SI words presented auditorily on all 19 EEG channels, hoping to see which regions of the brain are involved in this process. To complete this task, we used WinEEG software developed by Mitsar. The t-Student criterion was used with Tukey correction for the multiplicity of comparisons. In Figure 3, one can see ERP on two stimuli groups (explicit SI words and non-SI words) in auditory modality.

The black line reflects the brain’s response to explicit SI words, while the brown line reflects its response to non-SI words. The green line reflects the difference between the second and the first ERP. Each graph refers to a certain channel according to the 10–20 system. There are three lines under each graph. Each of the lines is marked with serifs of different sizes corresponding to the significance of the deviation from zero on the graph in a given time interval, according to t-Student and adjusted by Tukey for the multiplicity of comparisons. The significance of deviations from zero on the green graph means the reliability of the differences between the second and first ERPs. The time intervals in which ERP differ significantly are 360, 440, and 600 ms after the stimulus exposition. As we can see in the figure, the most significant differences in brain responses when processing explicit SI words and non-SI words are found in the central regions (premotor cortex).

At the next stage we carried out Repeated Measures Analysis of Variance sigma-restricted parameterization effective hypothesis decomposition for both modalities (visual, auditory) and three types of stimuli (explicit SI, implicit SI, and non-SI words) for Cz channel when all ERPs were averaged in time. The results are shown in Table 4.

The integrative activity of the brain significantly differs in the process of visual and auditory recognition of SI words depending on the modality, time parameters of brain activity, and iconicity.

In Figure 4, the influence of modality (visual, auditory) and iconicity on the time parameters in integrative activity of the brain is presented for Cz channel.

As Figure 4 shows, there are distinguishable differences between visual and auditory modalities, as well as three types of stimuli and time parameters of brain activity at Cz channel. Auditorily presented SI words evoke the P300 and the N400; implicit SI words evoke the P300 component more distinguishably, while explicit SI words evoke the N400 component more prominently. However, when those type of stimuli are presented visually, early components of ERP are only registered.

Figure 5 presents the results obtained for both modalities (visual, auditory) at Cz channel with a precise time window of 375–425 ms.

As Figure 5 shows, the most substantial differences between visual and auditory modalities for Cz channel and time window of 375–425 ms are recorded for auditory modality and explicit SI words. It appears that explicit SI words evoke larger long-latency ERP when presented auditorily than other types of lexical stimuli.

## 4. Discussion

Our first hypothesis concerning distinguishable differences in integrative brain activity while recognizing auditorily explicit iconic words, implicit iconic words, and non-iconic words was confirmed, our second hypothesis focusing on visual presentation of stimuli was not confirmed, and our third hypothesis concerning cross-modal correspondence involving premotor cortex while recognizing explicit iconic words was confirmed only for auditory modality.

Analyzing the obtained results, it is important to emphasize that the patterns of integrative brain activity reflected in the early components of ERP did not differ significantly between visual and auditory modalities. It is understandable because the early ERP components in the range 100–200 ms are known to be sensitive to lexical frequency in the process of sematic processing [34], which was similar for lexical stimuli in both modalities in our experimental design. Moreover, it is known that the N250 ERP components are sensitive to orthographic similarity [35] and the phonological status of the letters in the words [36], which again did not differ for auditory and visual modalities.

Surprisingly, the major differences detected in our study between both modalities were found for the late ERP components. This result mirrors previous studies wherein the P300 ERP component reflects the processes of distribution of arbitrary attention and stimulus categorization [37]. In the case of auditory recognition with the following visual stimuli categorization, those late positive complexes are much stronger for the auditory modality.

Other studies have shown that the late N400 ERP component is associated with lexical–semantic access [38]. In line with these studies, we observed particular differences between both modalities at the level of that indicator for explicit SI words processed auditorily. It is also worth mentioning that the N400 ERP was larger for semantically incongruent sentences than for semantically congruent ones [39]. In our case, incongruence could be caused by the interference between two streams of information encoded in explicit SI words presented auditorily. The first message is a semantic meaning of the word, which needs to be processed and recognized. The second one is an iconic message manifesting itself in the resemblance between the sound and the meaning of the word. In our case, all such words are verbs denoting movements. We assume that the obtained results may indicate the relative cognitive complexness of the experimental task of auditory recognition of SI words compared to that of visual recognition of the same stimuli. The reason for this is the following visual categorization, which supposedly involved auditory–visual cross-modal multisensory integration process.

Our further analysis established that the first result revealing statistically significant differences for auditorily presented explicit SI words in contrast to implicit SI words in the N400 ERP component corresponds to the fact that the N400 is larger for figurative language [40]. The question then arises as to why such significant differences were obtained for explicit SI words but not for the implicit ones. The answer can be found by analyzing the type of explicit and implicit SI words. It should be noted that all explicit SI words belong to a group of words depicting actions—howl, squeak, clap, sneeze, smack—while almost all explicit SI words refer to static objects—beetle, goose, itch, fluff, boar (except itch).

Thus, hypothetically taking into consideration the specifics of the experimental task, the process of recognition and the following categorization of the explicit SI words could involve audial and sensorimotor cross-modal multisensory integration, which was reflected in the larger N400 registered from the central channels (specifically Cz). Auditory recognition and the following visual categorization of the explicit SI words referring to static objects supposedly involved the process of cross-modal auditory-visual multisensory integration, which was reflected in the larger P300 component.

In this regard, it is quite possible that the resulting late ERP component N400 reflects not only the process of semantic processing of a linguistic stimulus, but also the component associated with the action symbolically manifested in this lexical stimulus. Similar results were obtained in earlier works. Thus, for the first time this component was discovered in response to the perception of semantically abnormal sentence endings in linguistic paradigms [41] and was associated with the semantic integration of the stimulus into the previous context.

In linguistics, the N400 is a reliable electrophysiological marker of semantic processing. Although its latency remains relatively constant [40], it has been shown that the amplitude of the N400 is sensitive not only to the degree of semantic inconsistency, but also to a number of other factors. For example, classical studies have shown that low-frequency words cause a greater amplitude for the N400 ERP component than high-frequency words [42]. In addition, word-like sequences of letters (or quasi-words) have also been shown to increase the amplitude of the N400 compared to actual words [43]. However, similar effects have also been observed in response to action-related stimuli. For example, pseudo-actions have been shown to modulate the amplitude of the N400 component similarly to pseudo-words [44]. In addition, it was shown that the N400 caused by the action resembles the linguistic N400 ERP component in shape and time, which indicates a functional similarity between both potentials [45].

Moreover, an interesting assumption was made that the N400 ERP component would reflect the process of semantic unification and sensorimotor integration initiated by a neural network consisting of storage (middle/upper temporal gyrus), a multimodal region (lower frontal gyrus), and a control area (dorsolateral prefrontal cortex), with the contribution of parietal zones [46]. It is likely that since our explicit SI words belonged to a group of words depicting an action, their auditory perception activated the process of sensorimotor integration through cross-modal correspondences; this was reflected in the more prominent appearance of the N400 ERP component compared to implicit SI words, referring mainly to the static objects.

Furthermore, in our case statistically reliable data were obtained on the appearance of a larger component of the P300 in the process of explicit SI words compared to non-SI words, which can be interpreted from the perspective of the distribution of arbitrary attention and categorization of lexical stimuli; however, it may also reflect the process of cross-modal correspondence. Similar data were obtained in a study aimed at registering ERP in the process of how people learn foreign sound–symbolic words in a state of congruence (the word and its translation coincide) and in a state of non-congruence (the translation does not correspond to the word) [20]. This second result lets us speculate that explicit SI words demand more cognitive resources in process of auditory recognition than non-SI words due to supposed cross-modal multisensory integration and additional cognitive load; this finding stems from the fact that the iconic message of those words is related to movements.

Special attention should be paid to the results obtained with the auditory presentation of explicit iconic words and the N400 registered from the central channels, which may indicate the process of cross-modal interaction during perception of this type of stimuli. The N400 ERP component detected at the central leads during the process of their auditory recognition of explicit stimuli can possibly be interpreted as an indicator of the process of cross-modal correspondence. We speculate that this might be due to the integrative role of the N400 in cognitive information processing related to motor tasks [47] and its predictive coding mechanisms when performing motor tasks [48]. In our case, all explicit iconic words are linked to movements in their meaning. Therefore, we assume that the results obtained may illustrate multisensory plasticity and cross-modal correspondence during processing of this kind of stimuli. However, we shall be careful with the interpretation because scalp electrode position is not equivalent to the cortical regions’ location. Further investigation and verification is needed to clarify which particular regions of the brain are involved in the process of auditory processing of explicit SI words.

## 5. Limitations

One significant limitation of our study is the fact that the research was carried out on stimuli material in the Russian language and the participants were all native speakers. Ideally, there should have been another experimental group of native English speakers and corresponding stimuli material with the same experimental design. It is a limitation because, according to our previous results, speakers of two typologically different languages, such as Russian and English, are not equally sensitive to iconicity [15].

Another limitation is a methodological one. It should be noted that although EEG studies can provide excellent temporal information, they lack sufficient spatial resolution to identify anatomical sources of activity in the brain during the process of processing iconic words [48]. In addition, it is questionable if the detected N400 reflects the process of cross-modal correspondence, which was only registered for explicit lexical stimuli presented auditorily. It should be noted that although the mechanisms used to interpret cross-modal correspondences may be informative, we should be careful when extending them to symbolic associations [49]. They largely depend on the type of stimuli presented, while the EEG data do not allow us to draw conclusions concerning the precise source of cortical activation.

Moreover, it was shown that performing tasks in an intermittent environment could be better than completing them in a completely silent room since humans are always confronted with noise [50]. Unfortunately, we did not take this issue into account in our current study but will consider it in future research.

## 6. Conclusions

We assume that the obtained results indicate a specific brain response associated with directed attention in the process of cognitive decision making tasks regarding explicit and implicit SI words presented auditorily. This finding may reflect a higher level of cognitive complexity involved in the identification of these types of stimuli. Supposedly, the process of audial recognition of explicit SI words involved the process of cross-modal correspondence (due to the fact that all these stimuli signify movements) and auditory–visual integration (due to the specifics of the experimental task). The absence of significant differences in the integrative brain activity in the process of visual recognition of explicit SI, implicit SI, and non-SI words may also be interpreted from the point of view of the experimental task specifics. It is probably due to the fact that this process involved visual modality exclusively and, thus, the cognitive complexness of the task was reduced. We may surmise that the process of visual recognition of iconic words did not affect the processes of multisensory integration and cross-modal correspondence.

## Figures and Tables

**Figure 1 brainsci-13-00681-f001:**
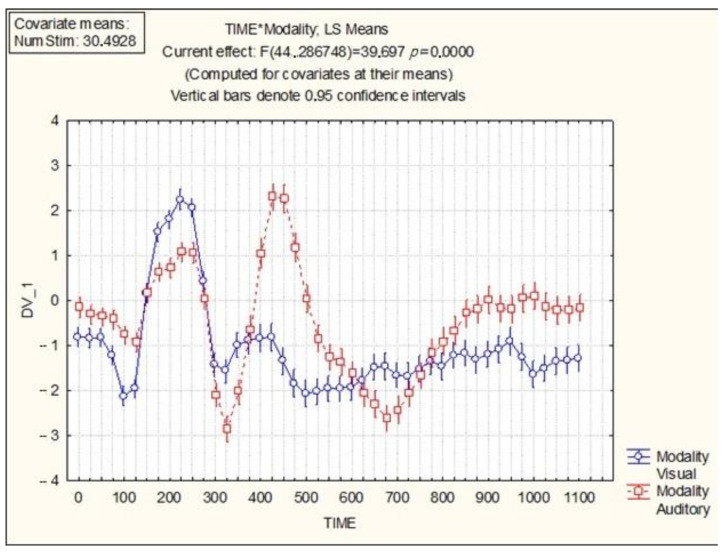
Modality (visual, auditory) impact on the time parameters in integrative brain activity. ‘*’ is a multiplication sign.

**Figure 2 brainsci-13-00681-f002:**
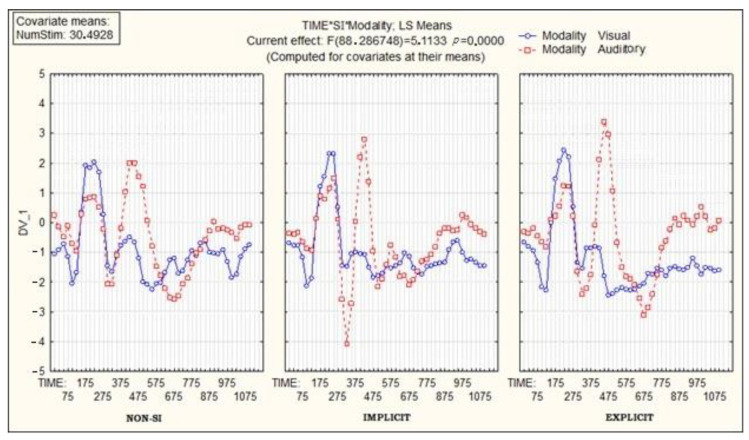
Differences between three groups of stimuli (explicit and implicit SI words and non-SI words) in both modalities (visual and auditory). ‘*’ is a multiplication sign.

**Figure 3 brainsci-13-00681-f003:**
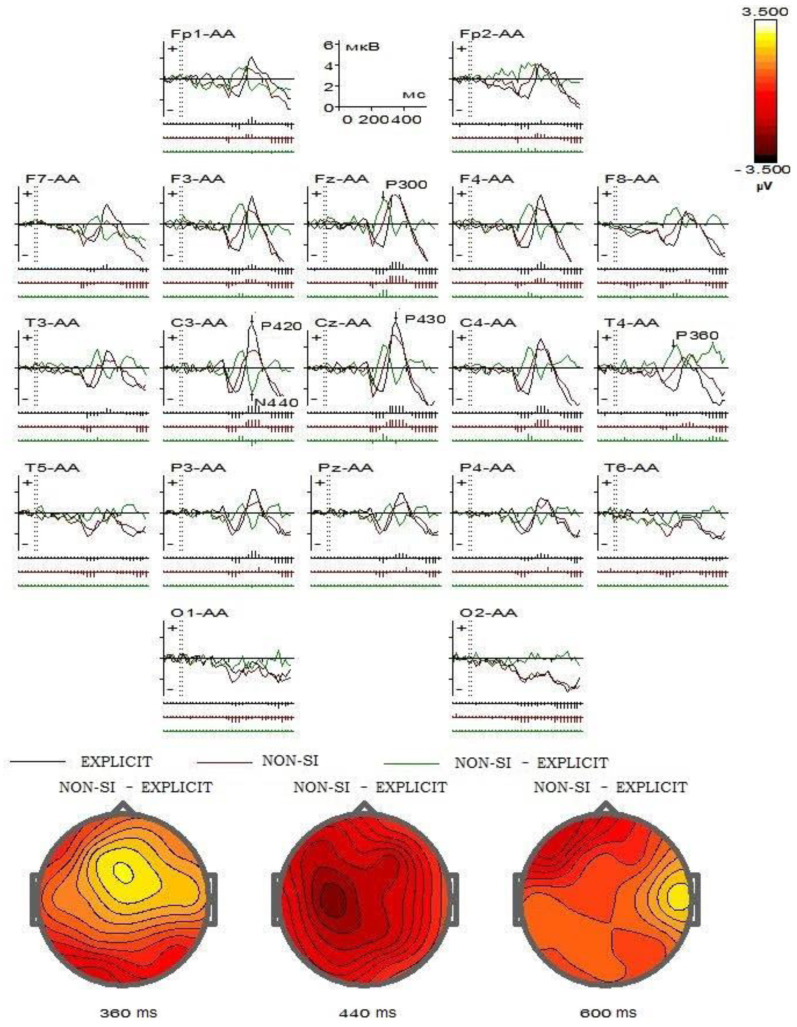
ERP on explicit SI words and non-SI words in auditory modality for all 19 channels. Colorful figure below shows topograms of differences on surface of head in microvolts according to the color scale in upper right corner of figure for three time frames of 360, 440, and 600 ms.

**Figure 4 brainsci-13-00681-f004:**
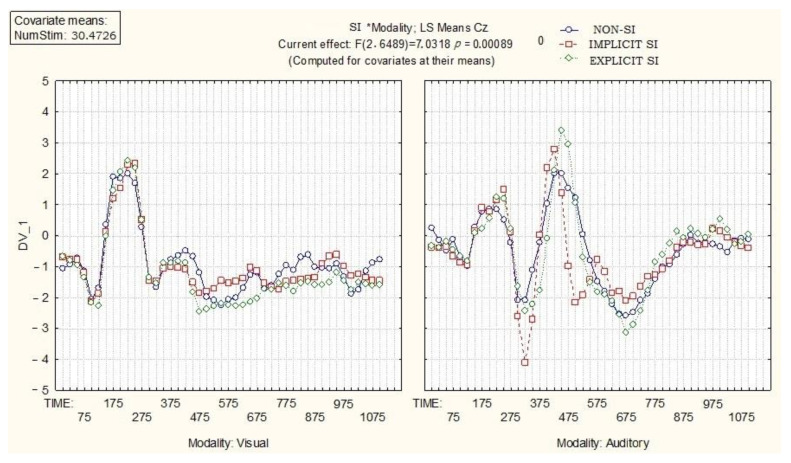
Differences between three groups of stimuli (explicit and implicit SI words and non-SI words) in both modalities (visual and auditory) for Cz channel. ‘*’ is a multiplication sign.

**Figure 5 brainsci-13-00681-f005:**
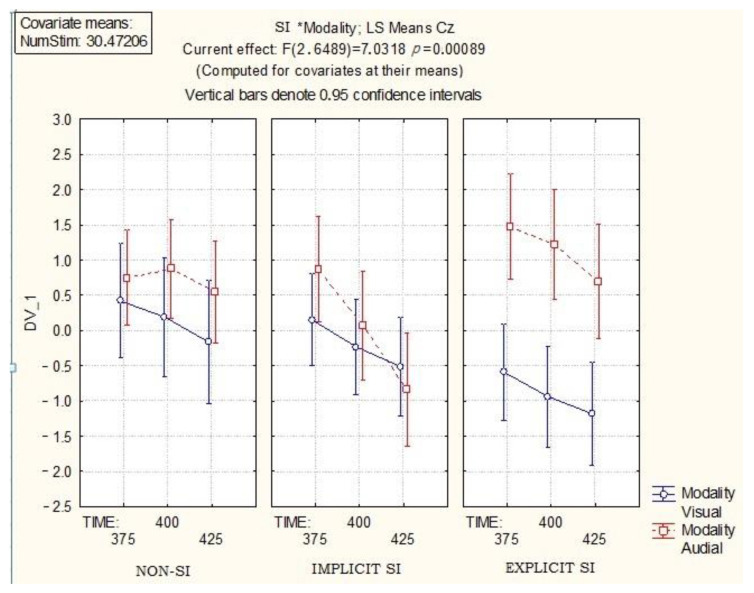
Differences between three groups of stimuli (explicit and implicit SI words and non-SI words) in both modalities (visual and auditory) for Cz channel for time window of 375–425 ms. ‘*’ is a multiplication sign.

**Table 1 brainsci-13-00681-t001:** Lexical stimuli for EEG experiment.

Explicit SI Words	Implicit SI Words	Non-SI Words
xlop (clap), čmok (smack), voj (howl), pisk (squeak), čix (sneeze)	žuk (bug, beetle), zud (itch), pux (fluff), xrjak (boar), gus’ (goose)	vosk (wax), svod (vault), syp’ (rash), taz (basin, bowl), trost’ (cane)

**Table 2 brainsci-13-00681-t002:** Repeated Measures Analysis of Variance results.

Effective Hypothesis Decomposition	SS	Degr. of Freedom	MS	F	*p*
Intercept	721	1	721.183	1.75397	0.185425
NumStim	350	1	350.271	0.85188	0.356054
SI	1669	2	834.493	2.02954	0.131479
Modality	8411	1	8411.299	20.45684	0.000006
SI × Modality	5783	2	2891.303	7.03184	0.000890
Error	2,668,101	6489	411.173		
LEADS	214	1	213.601	2.41226	0.120437
LEADS × NumStim	10	1	9.865	0.11141	0.738561
LEADS × SI	44	2	22.082	0.24938	0.779287
LEADS × Modality	7	1	7.175	0.08103	0.775916
LEADS × SI × Modality	159	2	79.574	0.89865	0.407169
Error	574,586	6489	88.548		
TIME	1215	2	607.604	27.79621	0.000000
TIME × NumStim	335	2	167.649	7.66950	0.000469
TIME × SI	684	4	171.070	7.82600	0.000003
TIME × Modality	171	2	85.321	3.90322	0.020201
TIME × SI × Modality	538	4	134.407	6.14875	0.000061
Error	283,689	12,978	21.859		
LEADS × TIME	83	2	41.559	2.45856	0.085598
LEADS × TIME × NumStim	80	2	39.919	2.36154	0.094315
LEADS × TIME × SI	99	4	24.825	1.46862	0.208789
LEADS × TIME × Modality	115	2	57.524	3.40297	0.033304
Error	219,379	12,978	16.904		

**Table 3 brainsci-13-00681-t003:** The Post Hoc analysis with Bonferroni correction results.

			Here Are the Same Conditions as in the Horizontal Lines					
	**SI**	**Modality**	1	2	3	4	5	6
1	Non-SI	Visual		1.000000	1.000000	1.000000	0.069497	0.182169
2	Non-SI	Audial	1.000000		0.041797	0.043261	0.000016	1.000000
3	Implicit	Visual	1.000000	0.041797		1.000000	0.689997	0.000523
4	Implicit	Audial	1.000000	0.043261	1.000000		0217682	0.014050
5	Explicit	Visual	0.069497	0.000016	0.689997	0.217682		0.000000
6	Explicit	Audial	0.182169	1.000000	0.000523	0.014050	0.000000	

**Table 4 brainsci-13-00681-t004:** Repeated Measures Analysis of Variance results for ERP averaged in time for Cz channel.

Effective Hypothesis Decomposition	SS	Degr. of Freedom	MS	F	*p*
Intercept	39,276	1	39,276.42	75.4776	0
NumStim	14	1	14.38	0.02763	0.867995
SI	830	2	415.2	0.79789	0.45032
Modality	20,869	1	20,895.67	40.15527	0
SI × Modality	2146	2	1072.95	2.0619	0.127295
Error	3,391,263	6517	520.37		
TIME	76,509	44	1738.84	31.78034	0
TIME × NumStim	3961	44	90.02	1.64527	0.004474
TIME × SI	20,178	88	229.3	4.19088	0
TIME × Modality	95,566	44	2171.96	39.69654	0
TIME × SI × Modality	6742	44	153.23	2.80049	0
Error	15,689,182	286,748	54.71		

## Data Availability

The data supporting reported results can be found at: [https://drive.google.com/drive/u/1/folders/1nDKB8s1KPCkiXR6SrbxN49qZMyOta20M] (URL accessed on 18 January 2022).

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
