# Peer review of "Neural Indicators of Visual Andauditory Recognition of Imitative Words on Different De-Iconization Stages"

_brainsci, 2023, doi:10.3390/brainsci13040681_

Round 1
Reviewer 1 Report
Dear authors,
I enjoyed reading your manuscript. The investigation of specific brain responses associated with directed attention in the process of cognitive decision-making tasks regarding explicit and implicit SI words and the level of cognitive complexity of identifying this type of stimuli is an interesting subject. The manuscript is well-written and structured. Besides, it provides a robust discussion. However, there are a few points to take into account in order to develop the paper.
1. The abstract specifies the research domain, but not the contribution of the paper to the target discipline. You may clearly state what gap your research is filling.
2. On page 2 you stated that ‘these mechanisms contribute to the assimilation and processing of lexical meaning’, you may refer to Banaruee et al. (2023), in that paper psychological mechanisms have been fully reviewed.
Banaruee, H., Khatin-Zadeh, O., & Farsani, D. (2023). The challenge of psychological processes in language acquisition: A systematic review, Cogent Arts & Humanities, 10(1), DOI: 10.1080/23311983.2022.2157961
3. The paper needs a proofread by a native scholar. There are some ambiguous sentences and vague expressions.
Good luck
Author Response
Dear Reviwer,
Please see the attachment

Reviewer 2 Report
The manuscript from Tkacheva and colleagues reported an ERP study on the processing of visual or auditory words differing in the degree of iconicity. They observed no ERP differences for iconicity for the visual words, but reported an N400 and a P300 effect for auditory words for different pairwise comparisons.
The research question of this study is interesting and highly relevant to the field. I also appreciate the effort from the authors to collect data from a large sample (n = 110). However, I do not that the manuscript in its current version satisfy the standard for academic publications. I list here a few major problems:
-
As an ERP study, there is no hypothesis regarding the targeted ERP components at the end of the introduction section (e.g., which ERP components may show effect for which comparisons). Eventhough the authors have introduced a number of relevant studies in the intro.
-
The authors included single data points as a factor into the factorial ANOVA with post-hoc pairwise comparison with Bonferroni correction. However, here it is not clear whether the correction is based on the numbers of time points (or other repetitions?). Also, the authors reported significant differences for auditory words in the N400 and P3 windows, but through out the text there is no mentioning of which time points (e.g., 300-500 ms for N400) were being analyzed for these windows. More importantly, even this this approach (including single time points in ANOVA) is technically OK, but it not appropriate enough. It is better to either do statistics based on the averaged of the hypothesized brain regions; or, if no clear hypothesis is available regarding the components (time windows), do a cluster-based permutation test // fdr to decide the time windows (see Groppe et al., 2021).
-
The Figures are not structured well enough to deliver the main message. Comparison between visual and auditory domain is reducent, as the ERPs from both modalities will differ for sure. It is more appropriate to illustrate a figure with 2 panels, one showing difference for three iconicities for auditory, and the other for visual.
References:
Groppe, D. M., Urbach, T. P., & Kutas, M. (2011). Mass univariate analysis of event-related brain potentials/fields II: Simulation studies. Psychophysiology, 48(12), 1726-1737.
Author Response
Dear Reviwer,
Please see the attachment

Reviewer 3 Report
Dear authors,
The paper is well written. However, I'll let you know some considerations and suggestions.
First, I'm not sure if you were looking for some brain mapping (19 channels seem weak for that outcome), and if not, please specify which channel(s) was (were) targeted in the procedure. And why.
Again, in the figures, the results could be clearer on what channels or if all are used. Even though, are you showing the power spectrum (relative power, absolute power, etc.)? We need units to interpret your results. That's important because visual activation will obviously differ from auditory activation when comparing the brain area.
I agree, overall, with your limitations. However, I'd exclude the sentence "However, this was not feasible due to COVID 2019 pandemic and unavailability of participants, native speakers of English, for contact data collection using EEG". Maybe it was one of the reasons, but there are other strong reasons for that! So, excluding the sentence and acknowledging it was a limitation is better. That's fine.
I'd recommend to your limitations, a study demonstrating that performing in a sound insulation chamber can influence the outcome. In other words, in 1Domingos et al. (2021) study, it was demonstrated that performing tasks in an intermittent environment could be better than doing it in a completely silent room since humans are always confronted with noise daily.
1Domingos, C., da Silva Caldeira, H., Miranda, M., Melício, F., Rosa, A. C., & Pereira, J. G. (2021). The Influence of Noise in the Neurofeedback Training Sessions in Student Athletes. International Journal of Environmental Research and Public Health, 18(24), 13223.
Author Response
Dear Reviwer,
Please see the attachment

Round 2
Reviewer 2 Report
I am sorry but the manuscript after the revision doe not satisfy academic publication standard.
Author Response
Dear Reviever,
Thank you very much for your time.
Kind regards,
Liubov
Reviewer 3 Report
Based on the modifications, I've no other comments.
Author Response

(The authors gave the same response as above.)
